# Impacts of the COVID-19 Pandemic on a Rural Opioid Support Services Program

**DOI:** 10.3390/ijerph191811164

**Published:** 2022-09-06

**Authors:** Jayme E. Walters, Aubrey E. Jones, Aaron R. Brown, Dorothy Wallis

**Affiliations:** 1Department of Social Work, Utah State University, 0730 Old Main Hill, Logan, UT 84322, USA; 2College of Social Work, University of Kentucky, 619 Patterson Office Tower, Lexington, KY 40506, USA

**Keywords:** opioid, COVID-19, public health, rural health, Appalachia, community health

## Abstract

During 2020, Kentucky saw the third highest increase in overdose deaths in the U.S. Employment issues, inadequate housing, transportation problems, and childcare needs present barriers to accessing treatment in rural areas. These barriers and others (e.g., technology) arose during the pandemic negatively affecting individuals in recovery and service providers as they adjusted services to provide primarily telehealth and remote services. This study examines the impact of COVID-19 in its early stages on an opioid use disorder (OUD) support services program in a nonprofit located in rural eastern Kentucky, part of the central Appalachia region. A qualitative design was applied, employing semi-structured interviews in early fall 2020. Participants were associated with one OUD support services program, including service recipients, program coordinators, and business vendors. Guided by the Social Determinants of Health framework, two-cycle coding–descriptive coding and pattern coding–was utilized. Codes were sorted into three patterns: changes to daily life; financial impacts; and service access and provision. Overall, early stages of COVID-19 brought increased stress for individuals in recovery, as they were taking on more responsibility and navigating a changing environment. Coordinators were under pressure to provide services in a safe, timely manner. Vendors vocalized their struggles and successes related to finances. These findings can help organizations make realistic adjustments and policymakers set reasonable expectations and consider additional financial support.

## 1. Introduction

Since the beginning of the COVID-19 pandemic, opioid-related overdose deaths in the U.S., especially those involving illicitly manufactured fentanyls, have accelerated across the country, including in the South [1]. Nationwide in 2020, there were over 100,000 drug overdose deaths [2], with 2000 involving Kentuckians [3]. During 2020, Kentucky saw the third highest increase in drug overdose death rates, with a 49% year-over-year increase [3] compared to a nationwide increase of 29% [2]. Opioids were involved in 90% of overdose deaths in Kentucky in 2020, and 71% involved fentanyl [3].

Moreover, individuals with substance use disorders (SUDs), opioid use disorder (OUD) included, often live with other comorbidities (e.g., cardiovascular, pulmonary, metabolic diseases, and increased susceptibility to infections), which were risk factors for COVID-19 [4,5,6,7,8,9]. When examining electronic health records in the U.S., Wang and colleagues found that individuals with OUD were at significant risk (AOR  =  2.42 [2.247–2.607]) for developing COVID-19, which was higher than other SUD subtypes despite having a similar prevalence of known risk factors for COVID-19 [10].

Even before the pandemic, challenges related to capacity of organizations and availability of services delayed OUD treatment in many parts of the U.S., particularly in predominantly low-income rural areas such as Appalachia that are disproportionately affected by opioid misuse, addiction, and overdoses [11,12,13]. Barriers to accessing treatments for OUD include treatment-related stigma and social determinants of health [14]. Employment-related issues, inadequate housing, transportation problems, long distances to treatment facilities, and childcare needs present substantial barriers to accessing treatment for OUD, particularly among marginalized populations in rural areas [14,15,16]. Certain sub-groups, including women, individuals recently incarcerated, and BIPOC, are more likely to need assistance to address barriers to OUD treatment access, utilization, and recovery in rural areas [13,17]. The COVID-19 pandemic exacerbated barriers and other challenges (e.g., technology), negatively affecting individuals in recovery and service providers [18,19]. Thus, rural OUD treatment providers and support services were forced to make swift adjustments in programming to ensure access for clients [20].

The adjustments to virtual services and reduced in-person services for rural health and social service organizations were more convoluted and compounded by economic and infrastructure issues. Some rural communities still do not have broadband (about 30 percent) [21], and about a third of rural residents do not have adequate access to technology (i.e., computer with a webcam, smartphone) [22,23]. In rural Appalachian Kentucky, residents have some of the lowest levels of broadband access, income, and educational attainment, while having some of the highest poverty and persistent poverty rates in the country [24,25]. For these reasons and others (e.g., computer literacy) [26], telephone visits during COVID-19 for services that did not require an in-person component provided increased accessibility for service users who did not have reliable access to technology [27].

Other types of SUD/OUD services that require an in-person component (i.e., methadone dispensing; needle exchange/provision) took advantage of the federal government’s temporary guideline adjustments that allow for take-home methadone dispensing [28] and prescriptions over the phone for buprenorphine [29]. Nonetheless, the support services—the human connection between service users and service providers and/or their peers—are lost when telephone services become a primary form of care and the videoconferencing option is inaccessible. Thus, some service providers used hybrid models of delivery [26], while others in rural communities delivered the services to the service users’ locations to ensure harm reduction care and support [20]. Early analysis of changes in service provision policies and delivery during the COVID-19 pandemic indicated an increase in access to OUD treatment and support for rural residents especially [30]. Still more studies are needed to determine how the worldwide crisis has affected OUD treatment and support service providers and their service recipients. Thus, providing context to the rural Appalachian Kentucky region, the present qualitative study examines how one opioid support services program, its employees, partners, and service recipients were impacted by the COVID-19 pandemic. 

## 2. Methods

### 2.1. Design and Guiding Framework

Approved by the University of Tennessee (Protocol No.: 20-05853-XP, Approved 15 July 2020) and Utah State University (Protocol No.: 11360, Approved 27 July 2020), the present study was part of a comprehensive investigation that examined the effectiveness of an opioid support services program in a rural community in Appalachia. The project began in the summer 2020 as the COVID-19 pandemic was in the midst of wreaking havoc internationally, and as such, the research team added interview questions to the protocol to account for the ways in which the pandemic was impacting the program, as well as its employees, partners, and clients, resulting in the current qualitative study.

Considering the rural environment and health aspects of the program of interest, the guiding framework for the entire investigation was the Social Determinants of Health (SDOH), which recognizes that context and conditions in people’s lives impact their health [31]. SDOH considers five areas that affect health risks and outcomes: Economic Stability; Education Access and Quality; Health Care Access and Quality; Neighborhood and Built Environment; and Social and Community Context [31]. In the present study, the five SDOH areas were used to generate codes and patterns in the deductive, two-cycle coding procedure (described in Section 2.4).

### 2.2. Sample and Sample Recruitment

Participants for the present study were recruited from an opioid support services program, Kentucky Access to Recovery (KATR), from one nonprofit located in eastern Kentucky, which is part of central Appalachia. The KATR program serves individuals 18 years of age and older who are recovering from OUD [32]. The KATR program is designed to help people recovering from opioid addiction by providing individuals in the program with vouchers to pay for essential needs which will enable them to be successful in their recovery [32]. Essential needs, also referred to as qualifying services, include housing, transportation, childcare, clothing, medical care, and other similar basic needs.

Three types of participants took part in the study, including program coordinators (i.e., staff; *n* = 3); vendors (i.e., local for-profit businesses who provided services to KATR service recipients in exchange for vouchers such as clothing or car repair; *n* = 4); and service recipients (i.e., KATR program clients; *n* = 12). To solicit participation of program coordinators and vendors, email addresses were provided by the nonprofit, and emails were sent up to three times to schedule interviews. For recruitment of service recipients, the nonprofit generated a list of individuals with current contact information, and investigators randomly selected 50 individuals to contact about the study. Postcards were mailed to service recipients as a first point of contact, and then within a week of the postcard, a phone call was placed to schedule interviews. For participation in the study, service recipients were mailed $15 Walmart gift cards. The vendors and program coordinators were not incentivized for their participation.

### 2.3. Data Collection

In fall 2020, semi-structured interviews (see Appendix A) via phone and Zoom were employed with all participant types. Each participant group addressed topics related to the impact of the opioid support services program on the service users, community level, and organizational level. Interviews were recorded and transcribed by research assistants and a professional transcription service, Rev.com.

**Service Recipients**. Semi-structured interviews with service recipient interviews were each 15 to 20 min in length via phone. Interviews were conducted until three of the investigators agreed—via weekly reviews and team dialogue—that data saturation with this participant group had been achieved (in this case, after 12 interviews). 

**Program Coordinators****.** One investigator led interviews with the three program coordinators, which were about 50 min in duration via Zoom.

**Vendors.** Semi-structured interviews with four KATR vendors, averaging 15 min in length, were conducted by one investigator via phone and Zoom.

### 2.4. Data Analysis

Data analysis for the present study was conducted independently by two investigators. A deductive two-cycle coding approach was adopted to analyze transcripts based on Miles and colleagues [33]: descriptive coding as cycle one and pattern coding as cycle two. Descriptive coding appoints one word or a short phrase to larger portions of data, and in this study, these words or phrases are reflective of the SDOH areas. Then in the second step of pattern coding, those granular codes are organized into overarching patterns, which also mirror the SDOH framework [33]. After completing both steps, investigators compared their analyses and resolved any inconsistencies through dialogue.

## 3. Results

Table 1 provides the self-identified demographic data for the service recipients who participated in the study. In all, 12 service recipients participated with an average age of 42, race as white, and an even mix of males and females. Three service coordinators and four vendors participated in the study. Demographic information for the service coordinators and vendors was not collected. 

Focusing on the individual- and organizational-level impacts, the results are organized into three patterns: (1) Daily life adjustments which are related to how individuals in recovery, as well as support services providers, retuned their regular habits and processes due to COVID-19; (2) Service access and modifications in service provision from the individual and organization perspective; and (3) Financial impacts of the pandemic on individuals in recovery, service providers, and vendors.

### 3.1. Daily Life Adjustments

Most participants from all three groups—program coordinators, vendors, and service recipients—indicated that they approached life differently as the world began to learn more about COVID-19 in summer and fall 2020. Several participants across the groups shared that they were wearing masks per a mask mandate at the time and limiting contact with people outside of their family and work. A few service recipients noted that they were not “afraid to catch it,” but at least three other service recipients said that the pandemic increased their anxiety:

“That has been really hard. At first, I was listening to the news, everything, Dr. Phil, anything I could find on TV, I was watching it. I’m not just a mother now, I’m a teacher, I’m a mother and a father, a housewife, and now I do everything… Hand sanitizer, washing hands, mask on, you know a lot to do.”(Service Recipient 7)

In addition to health-related changes, at least one person from each participant group remarked about changes in work and life schedules and the difficulty in managing the societal shifts: “… It has definitely affected everybody in the community.” (Vendor 2).

### 3.2. Service Access and Modifications in Service Provisions

**Service Access.** Most service recipients did not note major disruptions in KATR services due to the pandemic. However, issues existed prior to and persisted through the pandemic related to few available vendors in some categories, such as dental care, forcing some service recipients to drive across the county (i.e., up to an hour) or to a neighboring county to access needed services. A few service recipients and program coordinators shared that the pandemic intensified this problem as some vendors for KATR services, like dentists, closed down or reduced hours, while other vendors (e.g., clothing, auto repair) adjusted their facilities to accommodate customers. Additionally, some service recipients indicated that there were other services (e.g., group therapy; their children’s medical care) outside of KATR that they could not access, which caused stress and impacted their recovery. 

One of the coordinators shared that because KATR was a new program in the area, residents and the surrounding communities had little knowledge of its existence shortly prior to the onset of the pandemic. The massive economic crisis created by COVID-19 that impacted people from all backgrounds and incomes led to increased awareness of social service programs, including KATR. For people who likely needed the help even before COVID, they were finding their way to KATR:

“The good part is that we saw an increase in referrals. We have more clients. And, also the state of Kentucky offered more housing [benefits] during COVID…they upped it to $800 [from $400] which is a positive for our clients. So, we were able to help clients more even though it impacted them in such a way. It was kind of a good way because we were able to step in and be like, hey, we can help you with that…”(Coordinator 2)

**Service Provision.** Compared to service recipients, interviews with program coordinators yielded much more commentary regarding impacts of the COVID-19 pandemic on service provision. Swift and substantial changes in service provisions by KATR program coordinators and vendors became imperative, though for different reasons using different mechanisms. Focused on keeping service recipients in recovery during a very stressful time, program coordinators shifted service recipient intake and case coordination to mostly phone calls rather than in-person meetings, which presented challenges through which the coordinators had to navigate. Located in the Appalachian Mountains in a rural community, every coordinator shared how phone and internet services are unreliable and non-existent in some parts of the KATR service area:

“…your clients don’t have internet access to send the documentation they need to send and to prove who they are. And… it’s not just internet. We have very poor phone service…I struggle myself with phone service. I have to go stand outside on my porch in order to call people…”(Coordinator 3)

Also contributing to technology issues is the lack of technology (or knowledge to use it) to send the required documents via phone or computer, as some people did not have printers, scanners, fax machines, or even email addresses according to program coordinators. Coordinators acknowledged how much conducting services virtually slowed down the process and made it more difficult for all involved:

“…client comes into the office before all this happened…you had all your information right there in front of you, went to the copier, you make copies of it…. basically, you took care of everything right there really quickly…when you’re face to face, seems like they understand a little more. And then when you’re over the phone with them and tell them… they got kids in the background…something else going on in the home…. you really don’t have that personal one on one…. It’s different.”(Coordinator 1)

In addition to the technology challenges, the loss of in-person contact between service recipients and staff limited the staff’s abilities to informally assess the wellbeing of service recipients and provide encouragement and support as needed according to the program coordinators:

“I enjoyed interacting with the clients. When you are seeing someone face to face, it’s a totally different dynamic. You can gauge more from people that way… you lose a little bit of that on the phone…what’s going on in their life…”(Coordinator 3)

For service recipients who were not reliably reachable via phone or internet, program coordinators scheduled meeting times with service recipients when they were at other service providers’ offices (e.g., counseling) who would allow the service recipients to use their phones and/or computers. For others, program coordinators occasionally made home visits to obtain signatures and make important arrangements. Overall, though stressful and time-consuming, KATR staff were positive in their interviews about the quick adjustments made to ensure coordination services were being provided in the early days of the pandemic. From the perspective of service recipients during the early parts of the COVID-19 pandemic, all shared in some way in their interviews about the quality of services and how KATR program coordinators and their dedication to providing support while in OUD recovery:

“…being able to have somebody to fall back and talk to…that communication with the right person, the guidance that they can give you and help you with.”(Service Recipient 10)

### 3.3. Financial Impacts

The negative financial impacts of the COVID-19 pandemic occurred nearly immediately in rural eastern Kentucky, according to several study participants across groups. A few service recipients noted that they were already struggling prior to the pandemic, and the circumstances had not changed. However, two service recipients pointed out that the state of Kentucky provided temporary assistance, which helped considerably, though also may have had unintended consequences:

“… the diner I was working for pretty much shut the doors… so I was available to get the unemployment. I did lose food stamps because of that, but I’m doing okay with the unemployment…”(Service Recipient 6)

Vendors (i.e., the for-profit businesses that provide services to service recipients) were most vocal about the adverse financial consequences of the pandemic, sharing major struggles to stay afloat in the first few weeks and months of the worldwide crisis: 

“…it has affected everybody in the world, not just in the United States…on a business level, the first couple weeks…everybody was just really, really scared and kind of afraid to leave the house and it definitely impacted my business…after that, it kind of regulated back out and my business hasn’t suffered that much because of it…”(Vendor 2)

Other vendors too reported that after adjusting their service provision, increased online presence via social media, and the community becoming less fearful, their businesses were surviving again with varying losses and, for some, even thriving more than before COVID-19: 

“…I have done better than before COVID because the local people are supporting me more…We have really been amazed…”(Vendor 1)

## 4. Discussion

We explored how the COVID-19 pandemic impacted service delivery and service recipients in one rural program dedicated to recovery from OUD. An important distinction between the program and service recipients was that all service recipients must have been in active OUD recovery. The program itself is not a recovery program but, instead, seeks to provide the necessary supplemental assistance for those in recovery to help meet their basic needs and ensure recovery success from a holistic perspective. The findings from our study are situated within a rural Appalachian setting in eastern Kentucky. Rural communities, especially rural Appalachian communities, have a strong sense of community while maintaining their independence from those they perceive as outsiders (i.e., non-Appalachian people) and the study must be considered within this context. 

As with all of the U.S. and the world, the COVID-19 pandemic abruptly impacted daily life. This was no different for the participants included in our study. We found that study participants’ responses to the ways in which the COVID-19 pandemic has impacted them, their community, or the people they serve could be described in three main themes: daily life adjustments; service access and modifications in service provisions; and financial impacts.

Rural communities in central Appalachia face unique challenges regarding the geography, persistent poverty, and historical neglect of the region. The COVID-19 pandemic exacerbated these issues but created unique barriers to service provision within the region and for individuals in recovery from SUD—a population that has been recognized as increasingly vulnerable to health effects from COVID-19. The geography, poverty, and neglect of the region became exceedingly apparent in the interviews with service providers who discussed problems to continued service delivery due to the mountainous topography which exacerbated driving times for clients and service coordinators. Yet, the coordinators discussed that driving to the homes of service recipients was often the best way to maintain contact, make copies of important documents, and obtain signatures. This is partly due to the way the topography and regional neglect intersect. The regional neglect has led to increased rates of poverty [24,25] and has also led to the underdevelopment of common amenities found elsewhere in the U.S., such as broadband. Limited broadband, especially in the hollers of Kentucky, makes it nearly impossible for residents to have internet or rely on cellphone coverage [21,22,23,34].

In the present study, results indicated that program staff worked diligently to ensure that the service recipients—individuals who were in OUD recovery—received uninterrupted assistance needed to continue on their journeys successfully, which meant providing services via phone, computer, at appointments with other service providers, or at service users’ homes if necessary. Their efforts are reflective, in our opinion, of the community ties within this region—a strength that should be highlighted and is similar to other studies that share experiences and impacts of rural health and social service providers [35].

The implementation of different service delivery modes (i.e., hybrid) as demonstrated in this study paralleled other studies examining OUD/SUD services during the pandemic [20,26]. Program staff in the current investigation suggested that they had more participants after the onset of the pandemic than prior to it. This finding is consistent with previous studies that showed increases in rural program participants during the pandemic [30]. However, it is difficult to determine if increased access to services (i.e., multiple modes of service delivery—in person and telehealth) or increase in OUD/SUD—or both—is connected to the increase in rural program participants, and thus, future research should explore access in rural OUD service delivery and access further. 

Results of the present study were consistent with findings of studies pre-COVID [36] and after the onset of COVID [37] that note that rural Appalachian communities face technology challenges, particularly reduced internet access and quality of internet, when implementing telehealth services for physical and mental healthcare and social services. Access to broadband internet continues to grow with the investments of the federal government such as the Rural eConnectivity Program, among other development efforts (i.e., private businesses), but in a 2020 report, compared to urban areas with fewer than 2 percent lacking coverage, more than 20 percent of rural residents and over 27 percent of Tribal land residents in the U.S. still do not have fixed terrestrial 25/3 Mbps broadband [38]. Increasing access to and quality of technology and thereby, increasing access to and quality of telehealth services in rural communities will have profound positive impacts on rural residents’ wellbeing and health outcomes, especially in areas like central Appalachia that are dense with persistently poor counties, lack healthcare providers, and minimal transportation options [39]. Further, reducing barriers to healthcare by providing access to technology and more telehealth options for individuals in OUD recovery will increase the likelihood of continued success in recovery [40] and reduce health disparities in rural populations.

**Limitations.** Given that this study utilized qualitative research methods and evaluated a program in a specific region of eastern Kentucky, the results of this study should not be assumed to be generalizable to other geographical areas and with other populations. It should be noted, however, that the point of qualitative research is not generalizability; it is transferability [41]. 

Conducting research in rural communities can be challenging when discussing sensitive subjects, such as OUD recovery, which may have influenced service recipients’ and vendors’ agreement to participate or responses in the study. Further, our study is limited in that we did not collect demographic information for vendors or program coordinators; however, we did collect data on the service recipients. We also recognize COVID-19 and the historical neglect of the region as limitations to the study, both of which impacted our study design and who was able to participate in the study. Further, at the time of the study’s inception, all the researchers were Appalachian residents; however, none of the researchers were central Appalachian residents. Although the proximity of Tennessee and Kentucky are close, the regional differences are stark. It is possible this had an effect on who was willing to speak with the researchers and how the questions were answered given that we could be considered “outsiders.” Moreover, we, as researchers, recognize our roles and biases in the study (i.e., reflexivity). All researchers in this study consider themselves rural researchers with personal and/or professional ties to rural communities—particularly in the South—which may have influenced data collection and analysis.

## 5. Conclusions

Guided by the Social Determinants of Health framework, the present study sought to qualitatively examine the impact of the COVID-19 pandemic on a recently implemented opioid support services program in a rural Appalachian Kentucky region by interviewing those who were involved with the program. Based on interviews with 12 service recipients, three program coordinators, and four program vendors, the COVID-19 pandemic had significant individual-, organizational-, and community-level impacts on the implementation and management of the program. These impacts included adjustments to daily life during the pandemic, changes in access to services related to recovery participation, increased awareness of and participation in social service programs among community members, and sudden shifts in the ways services were provided. Barriers related to rural service provision such as transportation problems, childcare availability, and internet access persisted during the COVID-19 pandemic and in some cases were exacerbated by its impacts. The transition to virtual modes of service provision was difficult due to these barriers, and the loss of in-person contact between service recipients and program staff adversely affected rapport-building and assessments. Despite these impacts of the COVID-19 pandemic and due to the resilience and efforts of program staff, findings indicated that the program successfully provided needed support services to those recovering from OUD during a time when their continued participation in recovery was incredibly difficult. The findings from the present study may be useful for policymakers, public health professionals, and service providers in health and social services planning for the next public health crisis or emergency as it relates to rural service delivery in central Appalachia.

## Figures and Tables

**Table 1 ijerph-19-11164-t001:** This table provides demographic data for the service recipient participants of the sample. Demographic data are not available for the program coordinators, and vendors were not collected.

Service Recipient Demographic Characteristics (*n* = 12)		
Characteristic	*n*	*%*
Age	*M* (*SD*) = 41.92 (11.63)	
23–38	3	25.00%
39–46	5	41.67%
47–65	4	33.33%
Gender		
Male	6	50.00%
Female	6	50.00%
Race		
White	11	91.67%
Black	1	8.33%
Relationship Status		
Married/partnered	6	50.00%
Single	5	41.67%
Widowed	1	8.33%
Children		
Yes	11	91.67%
No	1	8.33%
Educational Attainment		
8th grade or less	1	8.33%
Some high school	2	16.67%
HS diploma or GED	3	25.00%
Some college	5	41.67%
Bachelor’s degree	1	8.33%
Employed		
Yes	7	58.33%
No	5	41.67%

## Data Availability

Not applicable.

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
