# Peer review of "Impacts of the COVID-19 Pandemic on a Rural Opioid Support Services Program"

_ijerph, 2022, doi:10.3390/ijerph191811164_

Round 1

Reviewer 1 Report

Extremely well written and interesting paper. I literally have no comments, and that's rare. I come from European Union so we don't face the same struggles when it comes to illicit drug abuse, as here we have to deal mostly with stimulants, and even though I am partially familiar with what you have to deal with in the United States, this article gave me one of the nicest backgrounds to the problem that I've ever seen. The study subject is increadibly important, as rural areas are often left forgotten by policy makers worldwide. Congratulations, I believe the article should be published in it's present form.

Reviewer 2 Report

The keywords need to be written according to PubMed MeSH subheadings, closely connected to the manuscript subject and the keyword Appalaciha should be omitted.

Referencing is wrongly done and needs to be changed according to instructions for authors.

The data from the Introduction and Background sections should be mixed into one much shorter Introduction section that will end with the clearly stated aim of this study.

Methods are poorly described and the study sample is too small even for a qualitative study. Authors' references to unpublished work must be omitted.

There are no data on the sociodemographic characteristics of study participants.

The data analysis tool is too weak for any stronger conclusion.

The author should explain the content of the semistructured interviews because readers need to know the themes of the conversations in order to better understand the answers of study participants.

The data presented within the results section are not enough for drawing an evidence-based conclusion.

In the Discussion section of the manuscript, the authors have stated that this study has serious flaws due to the fact that they did not collect any demographic data of their participants.

The discussion and conclusion sections are not based on the results of this study which are also poorly presented.

The generally known facts about telehealth and its influence on population health are not an issue for the discussion of this study.

Authors' contribution should be written according to the instruction of the authors.

The references section also needs to be rewritten according to the instructions to the authors.

Reviewer 3 Report

Thank you for your paper. However, the  study is more in the social than in the psychological area.

Reviewer 4 Report

Thank you for giving me the opportunity to read and comment a report “Impacts of the COVID-19 pandemic on a rural opioid support services program”, by Walters J.E., et al.

In the reviewed manuscript, the impact of the COVID-19 pandemic on rural residing individuals who are in recovery for OUD and seeking support services as well as on rural organizations who are providing services to individuals as part of their recovery has been investigated.

This is a potentially interesting report but at present it is not suitable for publication

Introduction and Background

·        The Background should be included in the introduction section, before describing the aim of the manuscript.

·        The length of this section is disproportionate.

·        The introduction should include, in summary form, the current state of the problem, the results of previous studies and research, as well as the aim of the study.

Methods

·        Which institution has approved the study?

·        It would be advisable for the authors to describe in more detail the content of the semi-structured interview.

Findings

·        According to the journal's recommendations, this section is called "Results”.

·        The Results section is too long and confusing. The authors could summarize the results and use some table or figure.

·        Results are not systematized. There are rather citations from questionaries.

Discussion

·        This section is not correctly structured. It should begin by summarizing the main findings of the research; it is not necessary to repeat the aim of the study. Subsequently, these results should be interpreted with respect to the aim of the study and the hypotheses proposed. In addition, the results should be compared with those obtained by other researchers. Finally, this section should include the limitations of the study.

·        This reviewer is missing a limitations of the study.

·        The low number of participants should be included as a limitation of the study.

Conclusion.

·        The "Conclusions" section is not correctly stated and should be rewritten. Authors should remember that conclusions must be derived exclusively from the results of the research.

·        In the opinion of this reviewer, the authors do not clearly conclude what was the impact of the COVID-19 pandemic on a rural opioid support services program.

Other minor issues:

·        Author affiliation is incomplete and not properly formatted.

·        According to the journal's instructions, references are incorporated in the manuscript using correlative numbers between square brackets.

·        Finally, it would be advisable to review the bibliography, since the references do not follow the format established by the journal.

Author Response

Please see the attachment,

Round 2

Reviewer 2 Report

There is a significant improvement in this manuscript and I congratulate to the authors on the work well done.

Appendix A helps with the understanding of the comments of interviewees.

It would be nice to have a table of sociodemographic data on service recipients. You only stated average age, race, and gender. It would be good to have a median and interquartile range of age, ethnicity, parenting, marital status, and educational level (and to state no answer for those who didn't answer).

Author Response

Thank you for your feedback and time in reviewing our manuscript. We have added a table for the demographic data of the service recipients. 

Reviewer 3 Report

Thank you for making several changes. However, still the theoretical part needs significant improvement with relevant studies. The findings do not bring relevant new information on a psychological level.

Author Response

Thank you for your feedback and time in reviewing our manuscript. We have made adjustments according to the other reviewers and editor, and we believe that our study contributes to the social and environmental challenges and successes experienced during the pandemic in a rural context.

Reviewer 4 Report

The manuscript has improved considerably and is now suitable for publication. 

I would like to add only one minor comment:

In the “Material and Methods” section, it would be desirable to include the protocol number and date of approval along with the institution that approved it.

Author Response

Thank you for your feedback and time in reviewing our manuscript. The protocol numbers and dates have been added.